# A Novel and Simplified Score for Determining Treatment Eligibility for Patients with Chronic Hepatitis B

**DOI:** 10.3390/v15030724

**Published:** 2023-03-10

**Authors:** Tanawat Geeratragool, Pisit Tangkijvanich, Supot Nimanong, Siwaporn Chainuvati, Phunchai Charatcharoenwitthaya, Tawesak Tanwandee, Watcharasak Chotiyaputta

**Affiliations:** 1Division of Gastroenterology, Department of Medicine, Faculty of Medicine Siriraj Hospital, Mahidol University, Bangkok 10700, Thailand; 2Department of Biochemistry, Faculty of Medicine, Chulalongkorn University, Bangkok 10330, Thailand; 3Center of Excellence in Hepatitis and Liver Cancer, Faculty of Medicine, Chulalongkorn University, Bangkok 10330, Thailand

**Keywords:** hepatitis b, treatment, score, simplification, Thailand

## Abstract

**Background:** International guidelines for hepatitis B infection (HBV) recommend initiating antiviral treatment based on viral replication with inflammation or fibrosis. HBV viral loads and liver fibrosis measurements are not widely available in resource-limited countries. **Aim:** To develop a novel scoring system for the initiation of antiviral treatment in HBV-infected patients. **Methods:** We examined 602 and 420 treatment-naïve, HBV mono-infected patients for derivation and validation cohorts. We performed regression analysis to identify parameters associated with the initiation of antiviral treatment based on the European Association for the Study of the Liver (EASL) guidelines. The novel score was developed based on these parameters. **Results:** The novel score (HePAA) was based on HBeAg (hepatitis B e-antigen), the platelet count, alanine transaminase, and albumin. The HePAA score showed excellent performance, with AUROC values of 0.926 (95% CI, 0.901–0.950) for the derivation cohort and 0.872 (95% CI, 0.833–0.910) for the validation cohort. The optimal cutoff was ≥3 points (sensitivity, 84.9%; specificity, 92.6%). The HePAA score performed better than the World Health Organization (WHO) criteria and the Risk Estimation for HCC in Chronic Hepatitis B (REACH-B) score, and it performed similarly to the Treatment Eligibility in Africa for HBV (TREAT-B) score. **Conclusions:** The HePAA scoring system is simple and accurate for chronic hepatitis B treatment eligibility in resource-limited countries.

## 1. Introduction

Chronic hepatitis B (CHB) infection is a major international health issue. In 2016, the estimated global prevalence of hepatitis B virus (HBV) infection was 3.9%, or 292 million infected patients. Almost 1 million HBV-infected patients die annually from liver-related complications, including cirrhosis and hepatocellular carcinoma (HCC) [1]. Early diagnosis and appropriate treatment can prevent this mortality. However, only 10.5% of HBV-infected persons were aware of their infection, with only 16.7% of that subgroup receiving treatment [2]. In 2016, the World Health Organization (WHO) established a strategy to eradicate viral hepatitis as a public health issue by 2030. This strategy aims to reduce infection by 90% and mortality by 65% [3,4]. Improving treatment coverage by scaling up and decentralizing treatment is essential to achieving this goal.

There are several international and local guidelines for CHB and HBV treatment, including the 2018 AASLD guideline (American Association for the Study of Liver Diseases) [5], the 2017 EASL guideline (European Association for the Study of the Liver) [6], the 2015 APSAL guideline (Asian-Pacific Association for the Study of Liver) [7], and the 2015 THASL guideline (Thai Association for the Study of the Liver) [8]. All of the mentioned guidelines recommend antiviral treatment in chronic HBV infection based on ongoing viral replication with significant inflammation or fibrosis. Ongoing viral replication can be evaluated via HBV DNA quantification, and significant fibrosis can be assessed with liver histology and liver stiffness measurements (LSMs). Unfortunately, these assessment tools are not widely available in resource-limited countries, such as many in Asia, including Thailand.

Many simplified criteria and scores have been developed to improve treatment coverage in countries where patients have difficulty obtaining complete evaluations before treatment. These scoring systems include the WHO simplified criteria and the Treatment Eligibility in Africa for Hepatitis B Virus (TREAT-B) scoring system. The WHO’s simplified criteria recommend initiating treatment based on persistently abnormal alanine transaminase (ALT) over 6–12 months, evidence of clinically defined cirrhosis, or an aspartate aminotransferase-to-platelet ratio index (APRI) > 2 [9]. The TREAT-B system recommends initiating treatment based on serum hepatitis B e antigen (HBeAg) and ALT levels [10]. These two systems do not require HBV DNA quantification or LSMs.

The WHO’s simplified criteria and the TREAT-B scoring system were validated in African, European, Australian, and Vietnamese populations. They showed variations in performance across geographic areas [10,11,12,13,14,15,16,17]. The variations were explained by differences in viral genotype, modes of HBV transmission, and rates of spontaneous loss of HBeAg [18,19]. Thus, a simplified score should be developed for each region.

This study aimed to develop and validate a novel, simple scoring system for the initiation of antiviral treatment in Thai HBV-infected patients. We compared the performance of our novel scoring system with other systems (WHO’s simplified criteria and the TREAT-B). The Risk Estimation for HCC in Chronic Hepatitis B (REACH-B) [20] scoring system was also used to validate the novel scoring system, given that the REACH-B is used to predict liver-related complications.

## 2. Materials and Methods

### 2.1. Study Population

This retrospective study drew upon 2 cohorts of treatment-naïve, chronic HBV mono-infected adult patients (aged ≥ 18 years) who attended the outpatient liver clinic and the internal medicine clinic. Patients for the derivation cohort were recruited from Siriraj Hospital, a large tertiary care center in Bangkok, Thailand, between January 2016 and December 2021. Patients for the validation cohort were sourced from King Chulalongkorn Memorial Hospital, another large tertiary care center in Bangkok, Thailand, between January 2020 and December 2021. We excluded patients with current or prior HBV antiviral treatment, co-infection with hepatitis C or human immunodeficiency virus, the presence of HCC, or pregnancy. In addition, patients missing any clinical or virological data needed for the development or evaluation of the proposed scoring system were not enrolled.

### 2.2. Data Collection

We collected data from the first patient visit or the visit occurring immediately before antiviral initiation at the outpatient liver clinic and internal medicine clinic. The data collected were related to demographic details; underlying diseases; HBV DNA quantification results of in-house reverse transcriptase polymerase chain reactions; HBeAg (using point-of-care testing); complete blood counts; 2 measurements of biochemical liver tests (total bilirubin, direct bilirubin, ALT, alanine transaminase, albumin, and globulin) that had been obtained more than 6 months apart; LSMs and controlled attenuation parameters (using FibroScan^®^, Echosens, Paris, France); liver ultrasonography; and liver histology.

### 2.3. Standard International Guidelines and Thai Guideline

We used the EASL Guidelines [6] as the gold standard to develop the novel scoring system in the derivation cohort. We also validated the novel scoring system with the AASLD guidelines [5], the APSAL guidelines [7], and the THASL guidelines [8] in both the derivation and validation cohorts.

The EASL guidelines [6] (grade of recommendation 1) recommend initiating antiviral therapy in patients who meet at least one of the following criteria:
HBeAg-positive or -negative patients with HBV DNA > 2000 IU/mL, ALT > upper limit normal (ULN), and/or at least moderate liver necroinflammation or fibrosis;Patients with HBV DNA > 20,000 IU/mL and ALT > 2 × ULN, regardless of the degree of fibrosis;Patients with cirrhosis and detectable HBV DNA.

    (*EASL guidelines defined ULN for ALT in healthy adults as 40 U/L*)

The AASLD [5] guidelines strongly recommend initiating antiviral therapy in patients who meet at least one of the following criteria:
Patients with high HBV DNA levels (>20,000 IU/mL if HBeAg-positive and >2000 IU/mL if HBeAg-negative) and elevated ALT levels >2 × ULN;Patient with cirrhosis with HBV DNA > 2000 IU/mL.

   (*AASLD defined the ULN for ALT in healthy adults as 30 U/L for males and 19 U/L for females.*)

The APASL [7] guidelines strongly recommend initiating antiviral therapy in patients who meet at least one of the following criteria:
Patients with high HBV DNA levels (>20,000 IU/mL if HBeAg-positive and >2000 IU/mL if HBeAg-negative) and persistently elevated ALT levels > 2 × ULN (at least 1 month between observations);Patients with high HBV DNA levels (>20,000 IU/mL if HBeAg-positive and >2000 IU/mL if HBeAg-negative) and a biopsy showing moderate-to-severe inflammation or significant fibrosis;Patients with decompensated cirrhosis and detectable HBV DNA;Patients with compensated cirrhosis and HBV DNA > 2000 IU/mL, even if their ALT levels are normal.

The THASL guidelines recommend antiviral therapy in patients who meet at least one of the following criteria:
Patients with high HBV DNA levels (>2000 IU/mL, regardless of their HBeAg status) and persistently elevated ALT levels > 2 × ULN (with at least 3 months between observations);Patients with high HBV DNA levels (>2000 IU/mL, regardless of their HBeAg status) and liver fibrosis ≥ F2 from LSM (defined as LSM ≥ 7.0 kPa) or biopsy shows moderate-to-severe inflammation or significant fibrosis;Patients with cirrhotic and detectable HBV DNA.

The antiviral treatment recommendations of the various guidelines are summarized in the Appendix A.

### 2.4. Simplified Criteria and Scoring Systems

The simplified criteria and scoring systems used for performance comparisons with the novel scoring system were as follows:

#### 2.4.1. WHO Simplified Criteria

The HBV treatment criteria provided by the WHO are for use with patients who do not have access to HBV DNA measurement in resource-limited countries. The criteria recommend initiating antiviral therapy in patients with (1) cirrhosis diagnosed by physical examination or an APRI > 2.0 or (2) persistently elevated ALT (at least twice over 6 months) [9].

#### 2.4.2. TREAT-B

The TREAT-B scoring system is based on categorized HBeAg status (negative, “0” points; positive, “1” point) and ALT score (<20 IU/L, “0” points; 20–39 IU/L, “1” point; 40–79 IU/L, “2” points; ≥80 IU/L, “3” points). From the original study, a cutoff of ≥2 points provided the best performance [10].

#### 2.4.3. REACH-B

The REACH-B scoring system assesses the risk of untreated patients developing HCC at 3, 5, and 10 years. REACH-B scores are derived from sex, age, ALT, HBeAg status, and HBV DNA measurement. The maximum score is 17 points [20]. We applied a score of 6/17 as the cutoff for a high risk of developing HCC and as suitable for antiviral initiation [10].

### 2.5. Rationale for Selecting Scoring System Parameters

We selected simple and widely available parameters that were previously verified as risk factors for liver-related complications following HBV infection. The parameters were age, sex, HBeAg, aspartate aminotransferase, ALT, albumin, total bilirubin, and platelet counts [5,6,7,18,20,21,22]. Unfortunately, we could not include some other core features (alcohol consumption and family history of HCC) because of the retrospective nature of our study.

### 2.6. Statistical Analysis

We developed the novel scoring system in 3 steps. First, univariable and multivariable logistic regression analyses identified independent parameters associated with the EASL criteria. A forward stepwise multivariable logistic regression analysis estimated regression coefficients, *β*, and *p*-values. Second, we developed a risk score model from the multivariable logistic regression model. Finally, we simplified the novel scoring system by adjusting constants and converting each parameter’s regression coefficient, *β*, to a simple integer point. The likelihood of meeting the EASL criteria for initiating antiviral therapy, based on the total points, was calculated using this equation [23]:(1)$$\frac{1}{1+\exp(-\sum_{i=0}^{p}\beta iWi-\beta \left(total\;points\right))}$$
where *βi* is the regression coefficient for the *i*th covariate, *Wi* is the reference value of the base category for the *i*th covariate, and *β* is the constant.

In both the derivation and validation cohorts, the performance of the novel scoring system was validated using the international guidelines (EASL, AASLD, and APASL) and the THASL guidelines as gold standards. Its performance was assessed via a receiver operating characteristic (ROC) curve, the area under the receiver operating characteristic curve (AUROC) [24], sensitivity, specificity, positive predictive value, negative predictive value, positive likelihood ratio, and negative likelihood ratio [25]. Sensitivity and 1–specificity were measured along the ROC curves’ vertical and horizontal axes, respectively. An optimal cutoff for the novel scoring system was selected to maximize the sum of sensitivity and specificity. We also compared the performance of the novel scoring system with other systems (the WHO simplified criteria and the TREAT-B and REACH-B scoring systems).

Statistical analyses were performed using STATA Statistical Software, release 14.2 (StataCorp LLC, College Station, TX, USA). Descriptive statistics were used to summarize the patients’ characteristics. Continuous data were presented as means and standard deviations or medians and IQRs (interquartile ranges), whereas categorical data were presented as frequencies and percentages. We compared continuous variables using Student’s *t*-tests and Mann–Whitney *U* tests and compared categorical variables using Pearson’s chi-squared tests and Fisher’s exact tests. We compared the quality of the AUROC using the roccomp test.

### 2.7. Ethics

The study protocol was approved by The Human Research Protection Program of the Faculty of Medicine at Siriraj Hospital, with the certificate of approval number Si745/2021.

## 3. Results

### 3.1. Participant Characteristics

A total of 1565 HBV-infected patients visited the outpatient liver clinic and internal medicine clinic of Siriraj Hospital between January 2016 and December 2021. For the derivation cohort, we excluded 855 patients due to acute HBV infection, human immunodeficiency virus or hepatitis C co-infection, the presence of HCC, or current or prior treatment for HBV infection. We also excluded 108 patients because data were missing. Eventually, 602 patients were included (Appendix A). Regarding the validation cohort, 420 treatment-naive CHB patients were enrolled.

The derivation and validation cohorts showed significant differences for many parameters: ALT, albumin, total bilirubin, platelet counts, APRI scores, and FIB−4 scores. Additionally, the participants in the derivation cohort were older and more predominantly male than those in the validation cohort. Last, according to the international guidelines (EASL, AASLD, and APASL) and the THASL guidelines, the participants in the derivation cohort had higher eligibility for antiviral treatment than those in the validation cohort (Table 1).

### 3.2. Development of the HePAA Scoring System

According to the univariable logistic regression analysis, the parameters associated with initiating antiviral therapy following EASL guidelines were male sex, HBeAg status, aspartate aminotransferase level, ALT level, albumin level, total bilirubin level, and platelet count (Table 2). Our forward stepwise multivariable logistic regression analysis determined that the independent parameters associated with the initiation of antiviral therapy were HBeAg status, platelet count, ALT level, and albumin level (*β* regression coefficients of 0.899, 0.097, −0.838, and 2.405, respectively; Table 2). From these regression coefficients, the logistic regression model we used was:
Score = (−1.604) + (0.899 × HBeAg) + (2.405 × Platelet < 150,000) + (0.097 × ALT) + (−0.838 × Alb)
where the value of HBeAg was “0” for negative and “1” for positive.

We developed a novel score called the “HePAA score” by adjusting the constant and converting the regression coefficient, *β*, to a simple integer point (Table 3). Individual HePAA scores ranged between 0 and 6 points, and they were calculated by summing the following:
HBeAg status (“0” for negative; “1” for positive)platelet count (“0” for ≥150 × 10^9^/L; “1” for <150 × 10^9^/L)ALT (“0” for <30 IU/L; “1” for 30–39 IU/L; “2” for 40–49 IU/L; “3” for ≥50 IU/L)albumin (“0” for ≥4 g/dL; “1” for <4 g/dL)

### 3.3. Performance of the HePAA Scoring System and Optimal Cutoff Selection

According to the EASL guidelines, the HePPA scoring system showed excellent performance for antiviral treatment eligibility, with an AUROC of 0.926 (95% CI, 0.901–0.950) in the derivation cohort. The sensitivities for total points of 1–6 were 96.7%, 92.1%, 84.9%, 50.0%, 16.4%, and 1.3%, respectively. The specificities for total points of 1–6 were 59.0%, 83.6%, 92.6%, 97.2%, 99.7%, and 100%, respectively (Appendix A). The optimal cutoff for the scoring system was selected by maximizing the sum of the sensitivity and specificity values. Thus, the optimal HePAA cutoff score for antiviral treatment eligibility was ≥3 points.

### 3.4. Validation and Comparison with Other Systems

The HePAA scoring system also showed excellent performance for antiviral treatment eligibility following EASL, with an AUROC of 0.872 (95% CI, 0.833–0.910). The performance of the novel scoring system was compared with the WHO simplified criteria and the TREAT-B and REACH-B systems. The HePAA scoring system performed better than the 3 other systems in the derivation and validation cohorts (Figure 1). The AUROCs for the HePAA scoring system were significantly higher than those for the simplified WHO criteria and the REACH-B scoring system (derivation cohort: *p* ≤ 0.001 and < 0.001, respectively; validation cohort: *p* ≤ 0.001 and < 0.001, respectively). Although the AUROCs for the HePAA scoring system were higher than those of the TREAT-B system, there were no statistically significant differences (derivation cohort: *p* = 0.550; validation cohort: *p* = 0.553). The sensitivities, specificities, positive predictive values, negative predictive values, positive likelihood ratios, and negative likelihood ratios of the HePAA scoring system and each of the 3 alternative systems are detailed in Table 4 (derivation cohort) and Table 5 (validation cohort). The HePAA system demonstrated similar performance when using all guidelines (AASLD, APASL, and THASL) as gold standards (Appendix A).

## 4. Discussion

The WHO has established a strategy to eliminate HBV infection by 2030. One of the related policies is that up to 90% of HBV-infected persons should be diagnosed, with at least 80% of those individuals being given HBV treatment. Unfortunately, many measurements that need to be evaluated before HBV treatment, such as HBV viral loads and liver fibrosis assessments, are not readily available to patients in resource-limited countries. A simplified antiviral treatment eligibility scoring system is the key to scaling up and decentralizing the treatment of HBV infections in such countries. The TREAT-B scoring system and the WHO simplified criteria are the only systems that were designed expressly for this purpose. The TREAT-B system was developed for HBV-infected patients who were all African, whereas the WHO criteria have only 2 parameters to be considered before deciding to treat patients. Therefore, neither system might be appropriate for application to Asian patients.

There are several scoring systems for predicting HCC development, such as the REACH-B [20], PAGE-B (platelets, age, gender-hepatitis B) [26], modified PAGE-B [27], THRI (Toronto HCC Risk Index) [28], and CU-HCC (Chinese University-Hepatocellular Carcinoma) [21]. These systems have demonstrated that racial differences present dissimilar independent risk factors for predicting HBV infection outcomes. Racial differences were also apparent in performance studies of the WHO simplified criteria and the TREAT-B scoring system conducted among African, European, Australian, and Vietnamese populations. The sensitivities of the WHO criteria and the TREAT-B scoring system for chronic hepatitis B treatment eligibility ranged between 53.0% and 100.0% and from 73.9% to 98.8% [10,11,12,13,14,15,16,17], respectively. Similarly, the specificity ranges of the WHO criteria and TREAT-B system for chronic hepatitis B treatment eligibility were from 40% to 77.4% and 57.5% to 88.0 [10,11,12,13,14,15,16,17], respectively.

The variations in sensitivity and specificity result from differences in viral genotype, the modes of HBV transmission, and the rate of spontaneous loss of HBeAg. The predominant HBV genotypes in Caucasian, Asian, and African populations were A/D/G, B/C, and D, respectively [29,30]. Vertical HBV transmission was the most common mode in Asia, whereas early horizontal transmission was the most common mode in Africa [18,31]. The rate of spontaneous loss of HBeAg in Africans was faster than that in Asians. In the second decade, 90% of HBV-infected Africans could clear HBeAg. Therefore, it is necessary to develop an HBV antiviral treatment eligibility scoring system targeting the specific population of each region.

The TREAT-B scoring system is based on HBeAg status and ALT level, whereas the HePAA system draws upon HBeAg status, platelet count, ALT level, and albumin level. The dissimilarities in the independent parameters may result from multiple factors. First, there were marked differences in the characteristics of the participants in the derivation cohorts used to develop the HePAA and TREAT-B scoring systems. The participants in the derivation cohort for the HePAA system were older and had higher HBV DNA levels, a more frequent presence of HBeAg, and a greater presence of significant fibrosis. However, the participants in the derivation cohort for the TREAT-B system had higher levels of transaminase.

Second, a small proportion of participants involved in the TREAT-B system development had significant fibrosis and cirrhosis (6% and 3%, respectively) [10]. Therefore, the TREAT-B scoring system did not include fibrosis parameters (albumin level and platelet count), unlike the HePAA system. Consequently, patients with advanced fibrosis and minimal ALT elevation might not meet the criteria for HBV treatment based on the TREAT-B system. Moreover, most patients in the TREAT-B derivation cohort had mild disease severity. The prevalence of antiviral-indicated, HBV-infected patients was only 7%, compared with 27.5% in our HePAA study and 32–64% in previous reports [1,32].

We selected an optimal cutoff of ≥3 points based on the highest sensitivity- and specificity values’ summation. However, the optimal cutoff may be modified to suit the local context. We found that a cutoff of ≥2 points improved sensitivity from 84.9% to 92.1% in the derivation cohort and from 87.9% to 97.8% in the validation cohort, albeit with decreased specificity. It is recognized that overtreatment can cause medical and financial burdens because of the life-long treatment involved, the need for monitoring, and possible adverse medication events. However, expanding treatment coverage can reduce liver-related complications, lower mortality rates, and improve quality of life. Further cost-effectiveness analysis using this HePAA score is required.

The HePAA scoring system was excellent at selecting patients for the initiation of antiviral therapy, according to the EASL guidelines, in both the derivation and validation cohorts. The HePAA system also performed better than the WHO simplified criteria and the REACH-B scoring system, and it performed similarly to the TREAT-B system. The international guidelines suggest initiating antiviral treatment in CHB-infected patients based on ongoing viral replication with significant inflammation or fibrosis. Unlike the TREAT-B system, the HePAA scoring system includes fibrosis-related parameters (albumin level and platelet count). Consequently, the HePAA system should perform better for patients with fibrosis than the TREAT-B system. However, the HePAA system did not outperform TREAT-B. The small number of patients with significant fibrosis in the HePAA derivation and validation cohorts could be an explanation.

Our study has several strengths. The HePAA scoring system is the first to be designed for an Asian population and shows excellent performance. Additionally, HePAA scores are calculated using blood tests widely used in resource-limited countries: HBeAg status, complete blood count, and liver biochemical tests (albumin level and ALT level). Consequently, this simple-to-use scoring system can be readily applied in various clinical settings. Moreover, the HePAA scoring system includes inflammatory and fibrosis parameters, which are part of the treatment indications specified by international guidelines. Finally, the HePAA system has excellent performance among different populations, as demonstrated during the validation phase of its development.

Our study also has some limitations. First, the participants do not represent the general population because the derivation and validation cohorts were drawn from patients at tertiary care centers. Second, due to its retrospective nature, we could not collect important predictive parameters (such as a family history of HCC) or information to exclude other causes of hepatitis (alcohol consumption and medication use). Finally, although the HePAA scoring system does not require HBV DNA measurement, it is still needed, especially for treatment-monitoring purposes.

## 5. Conclusions

In conclusion, because it is based on HBeAg, platelet count, ALT level, and albumin level, the HePAA scoring system is simple to use and can be readily applied in various clinical settings. It also demonstrates excellent performance in determining CHB treatment eligibility for an Asian population. This novel scoring system has the potential to facilitate HBV infection treatment coverage in resource-limited countries. However, external validation and cost-effectiveness analyses should be performed to determine the system’s degree of generalizability and its ability to be applied in real-world settings.

## Figures and Tables

**Figure 1 viruses-15-00724-f001:**
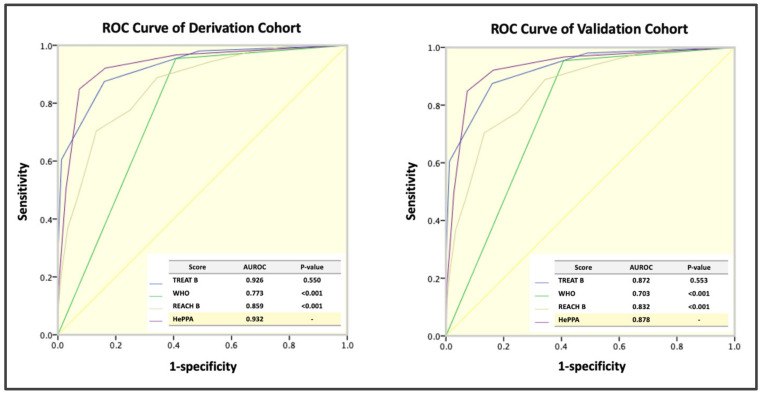
Receiver operating characteristic curves for the HePAA, TREAT-B, and REACH-B scoring systems and the WHO simplified criteria for indicating EASL treatment eligibility in the derivation (*n* = 602) and validation cohorts (*n* = 420). EASL, European Association for the Study of the Liver; REACH-B, Risk Estimation for Hepatocellular Carcinoma in Chronic Hepatitis B; TREAT-B, Treatment Eligibility in Africa for the Hepatitis B Virus; WHO, World Health Organization.

**Table 1 viruses-15-00724-t001:** Characteristics of study participants in the derivation cohort (*n* = 602) and validation cohort (*n* = 420).

	Derivation Cohort (*n* = 602)	Validation Cohort (*n* = 420)	*p*-Value **
Age (years)	51.10 ± 13.825	43.24 ± 11.97	<0.001
Male sex, *n* (%)	279 (46.3%)	227 (54.0%)	0.010
BMI (kg/m^2^)	23.39 ± 3.87	23.56 ± 3.68	0.480
HBV DNA <2000, *n* (%) 2000–20,000, *n* (%) >20,000, *n* (%)	283 (47%) 88 (14.6%) 231 (38.4%)	174 (41.4%) 67 (16.0%) 179 (42.6%)	0.697
HBeAg positive, *n* (%)	88 (14.6%)	58 (13.8%)	0.697
Liver fibrosis by LSM, *n* (%) F0–1 (LSM < 7.9) F2–3 (LSM 7.9–9.4) F4 (LSM > 9.4)	334 (83.5%) 22 (5.5%) 44 (11%)	297 (83.9%) 21 (5.9%) 36 (10.2%)	0.911
AST (IU/L), median (IQR)	25.0 (20.0, 40.0)	26.0 (20.0, 36.0)	0.589
ALT (IU/L), median (IQR)	25.0 (17.0, 47.0)	30.0 (20.0, 50.0)	<0.001
Albumin (g/L), mean ± SD	4.26 ± 0.47	4.48 ± 0.52	<0.001
TB (IU/L), median (IQR)	0.50 (0.40, 0.70)	0.8 (0.1, 1.0)	<0.001
Platelet (10^9^/L), median (IQR)	242 (201, 279)	231 (199, 270)	0.018
APRI, median (IQR)	0.269 (0.192, 0.453)	0.390 (0.270, 0.560)	<0.001
FIB−4, median (IQR)	1112 (0.789, 1.594)	1146 (0.642, 1.219)	<0.001
Eligible for EASL treatment criteria, *n* (%)	165 (27.5%)	92 (21.9%)	0.042
Eligible for AASLD treatment criteria, *n* (%)	170 (28.2%)	94 (22.4%)	0.032
Eligible for APASL treatment criteria, *n* (%)	162 (27.2%)	94 (22.4%)	0.079
Eligible for THASL treatment criteria, *n* (%)	171 (28.4%)	94 (22.4%)	0.028

AASLD, American Association for the Study of Liver Diseases; ALT, alanine aminotransferase; APASL, Asian Pacific Association for the Study of the Liver; APRI, AST-to-platelet ratio index; AST, aspartate aminotransferase; BMI, body mass index; EASL, European Association for the Study of the Liver; FIB-4, fibrosis 4; HBeAg, hepatitis B e-antigen; HBV, hepatitis B virus; IQR, interquartile range; REACH-B, Risk Estimation for Hepatocellular Carcinoma in Chronic Hepatitis B; WHO, World Health Organization. ** *p*-values were obtained using Student’s *t*-tests for normally distributed continuous variables, Mann–Whitney *U* tests for nonnormally distributed continuous variables, and chi-squared tests for categorical variables.

**Table 2 viruses-15-00724-t002:** Predictors for treatment eligibility in the derivation cohort (*n* = 602).

Variables	Not Eligible for Treatment (*n* = 602)	Eligible for Treatment by EASL Guidelines (*n* = 172)	*p*-Value	Final Model Selected by Forward Stepwise Regression	
				*β* Regression Coefficient	*p*-Value
Age, mean ± SD (years)	50.98 ± 13.62	51.47 ± 14.39	0.695		
BMI, mean ± SD (kg/m^2^)	23.37 ± 3.92	23.42 ± 3.78	0.901		
Male sex, *n* (%)	178 (40.70)	101 (61.20)	<0.01		
HBeAg, *n* (%)	29 (6.60)	59 (35.8)	<0.01	0.899	0.036
AST, median (P_25_, P_75_) (IU/L)	22 (19, 27)	58 (42, 87)	<0.01		
ALT, median (P_25_, P_75_) (IU/L)	20 (15, 28)	80 (47, 123)	<0.01	0.097	<0.01
Albumin, mean ± SD (g/L)	4.34 ± 0.41	4.11 ± 0.55	0.015	−0.838	0.025
TB median (P_25_, P_75_) (IU/L)	0.48 (0.37, 0.64)	0.64 (0.46, 0.90)	<0.01		
Platelet < 150 × 10^9^/L, *n* (%)	7 (1.6%)	24 (14.5%)	<0.01	2.405	<0.01

ALT, alanine aminotransferase; AST, aspartate aminotransferase; BMI, body mass index; EASL, European Association for the Study of the Liver; HBeAg, hepatitis B e-antigen; TB, total bilirubin. *p*-values were obtained using Student’s *t*-tests for normally distributed continuous variables, Mann–Whitney *U* tests for nonnormally distributed continuous variables, and chi-squared tests for categorical variables.

**Table 3 viruses-15-00724-t003:** Development of HePAA scoring system based on the logistic regression model.

Predictors	Categories	*β* Regression Coefficient	Adjusted with Constant **	Points
HBeAg	Negative	0	0	0
	Positive	1095	4.568	1
Platelet (10^9^/L)	<1,500,000	2.2561	5.928	1
	≥1,500,000	0	0	0
ALT (IU/L)	<30	0	0	0
	30–39	1558	5093	1
	40–49	3143	6754	2
	≥50	4757	8361	3
Albumin (g/L)	<4	0.436	4236	1
	≥4	0	0	0

ALT, alanine aminotransferase; HBeAg, hepatitis B e-antigen. ** Constant = −3399.

**Table 4 viruses-15-00724-t004:** The performance of various systems in determining chronic hepatitis B treatment eligibility, according to EASL guidelines, in the derivation cohort (*n* = 602).

	TREAT-B (≥2)	Simplified WHO Criteria	REACH-B (≥6)	HePAA (≥2)	HePAA (≥3)
AUROC	0.926 (0.901–0.950)	0.773 (0.742–0.805)	0.859 (0.825–0.893)	0.932 (0.907–0.957)	
*p*-Value **	0.550	<0.001	<0.001	N/A	
Sensitivity (%)	86.7 (80.5–91.5)	94.5 (89.9–97.5)	98.2 (94.8–99.6)	92.1 (86.6–95.9)	84.9 (78.2–90.2)
Specificity (%)	86.3 (82.7–89.4)	62.0 (57.3–66.6)	37.5 (33.0–42.3)	83.6 (79.2–87.5)	92.6 (89.2–95.2)
Positive Predictive Value (%)	70.4 (63.7–76.6)	48.4 (42.9–54.1)	37.2 (32.7–42.0)	72.5 (65.7–78.7)	84.3 (84.3–77.6)
Negative Predictive Value (%)	94.5 (91.8–96.5)	96.8 (94.0–98.5)	98.2 (94.8–99.6)	95.8 (92.7–97.8)	92.9 (89.5–95.4)
Positive Likelihood Ratio	6.31 (4.95–8.04)	2.49 (2.20–2.82)	1.57 (1.46–1.70)	5.63 (4.38–7.23)	11.46 (7.75–16.94)
Negative Likelihood Ratio	0.15 (0.10–0.23)	0.09 (0.05–0.17)	0.05 (0.02–0.15)	0.09 (0.05–0.16)	0.16 (0.11–0.24)

AUROC, the area under the receiver operating characteristic curve; EASL, European Association for the Study of the Liver; REACH-B, Risk Estimation for Hepatocellular Carcinoma in Chronic Hepatitis B; TREAT-B, Treatment Eligibility in Africa for the Hepatitis B Virus; WHO, World Health Organization. ** *p*-values were obtained using STATA’s “roccomp” command.

**Table 5 viruses-15-00724-t005:** The performance of various systems in determining chronic hepatitis B treatment eligibility, according to EASL guidelines, in the validation cohort (*n* = 420).

	TREAT-B (≥2)	Simplified WHO Criteria	REACH-B (≥6)	HePAA (≥2)	HePAA (≥3)
AUROC	0.872 (0.833–0.910)	0.703 (0.668–0.737)	0.832 (0.789–0.875)	0.878 (0.843–0.913)	
*p*-Value **	0.553	<0.001	<0.001	N/A	
Sensitivity (%)	84.8 (75.8–91.4)	95.7 (89.2–98.8)	93.5 (86.3–97.6)	97.8 (92.3–99.7)	87.9 (79.4–93.8)
Specificity (%)	72.0 (66.8–76.7)	55.2 (49.6–60.7)	36.0 (30.8–41.4)	56.9 (51.3–62.3)	73.7 (68.6–78.4)
Positive Predictive Value (%)	45.9 (38.2–53.7)	37.5 (34.5–40.5)	29.1 (23.9–34.6)	38.7 (35.7–41.8)	31.5 (27.4–35.9)
Negative Predictive Value (%)	94.4 (90.8–96.9)	97.8 (94.5–99.2)	95.2 (89.8–98.2)	98.9 (95.9–99.7)	97.8 (96.2–98.7)
Positive Likelihood Ratio	3.02 (2.49–3.67)	2.13 (1.88−2.43)	1.46 (1.32–1.61)	2.27 (2.00–2.58)	3.34 (2.75–4.07)
Negative Likelihood Ratio	0.21 (0.13–0.34)	0.08 (0.03–0.21)	0.18 (0.08–0.40)	0.04 (0.01–0.15)	0.16 (0.09–0.29)

AUROC, the area under the receiver operating characteristic curve; EASL, European Association for the Study of the Liver; REACH-B, Risk Estimation for Hepatocellular Carcinoma in Chronic Hepatitis B; TREAT-B, Treatment Eligibility in Africa for the Hepatitis B Virus; WHO, World Health Organization. ** *p*-values were obtained using STATA’s “roccomp” command.

## Data Availability

Not applicable.

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
