# Peer review of "A Novel and Simplified Score for Determining Treatment Eligibility for Patients with Chronic Hepatitis B"

_viruses, 2023, doi:10.3390/v15030724_

Round 1

Reviewer 1 Report

I consider this project to be solid and consistently implemented. The results can help doctors to make a decision for hepatitis B therapy because presented HePAA scoring system is easy. For publication.

Author Response

We thank the reviewer for taking the time to review our manuscript and providing helpful comments. We greatly appreciate the opportunity to revise and resubmit our manuscript to Viruses. We very much hope that our responses appropriately address his or her questions and concerns. 

Reviewer 2 Report

This study validates a simplified Score for determining treatment elegibility for patients with hepatitis B in resource-limited countries where DNA viral load determination and fibrosis assessment is not always available.

The score showed excellent performance, better than other simplified tests. In my opinion it can be very useful in clinical practice. 

I only have minor suggestions that I enclose.

Author Response

(The authors gave the same response as above.)

Reviewer 3 Report

Title: A Novel and Simplified Score for Determining Treatment Eligibility for Patients with Chronic Hepatitis B
Type: research article
General comment: In this work, it has been described the development and validation of a novel scoring system for the initiation of antiviral treatment in HBV-positive patients.
Comment 1. The work is interesting and well conducted. The scoring system appear to be reliable and robust as currently employed systems provided form various international agencies. Enclosed please see some specific comments alongside minor suggestions to further improve this work, which I recommend for publication in viruses MDPI.
Comment 2. The formatting citation of Viruses is [cit]. Please revise the work accordingly. Authors can check the following work https://www.mdpi.com/1999-4915/15/3/614
Comment 3. The study aim should be better detailed in the abstract. In the current form it is difficult to understand
Comment 4. Introduction “Among them are the 2018 AASLD …. the Study of the Liver).” English should be revised
Comment 5. Study groups’ numerosity and mean ages should be introduced in the 2.1 section
Comment 6. There is no mention of the etic committee authorization in the methods.
Commen7 . I suggest reorganizing section 2.3 in order to improve its readability.

Minor
1)    The aim should be better detailed in the abstract
2)    Abstract, Is this an abbreviation? “HePAA”, sdame question “ HBeAg
3)    Please include AUC values and P values in figure 1. I also suggest enlarging the words for a better visualization of the figure by readers.
4)    More references should be included in the methods. Author are encouraged to include these works in the statistical section of the methods PMID: 35124947 and https://www.ncbi.nlm.nih.gov/books/NBK557491/ and the following work for Pearson’s chi-squared test and Fisher’s exact test (PMID: 34970247)
5)    area under the receiver operating characteristic curve is usually known as AUC
6)    please include Thailand (as study country) near hospitals’ names.
7)    Table 1 column titles are misaligned.
8)    Despite normality tests are applied for table 1 data analyses, there is no mention of this test in the statistical section of the methods.

Author Response

(The authors gave the same response as above.)
